# Effect of High Energy Low Protein Diet on Lipid Metabolism and Inflammation in the Liver and Abdominal Adipose Tissue of Laying Hens

**DOI:** 10.3390/ani14081199

**Published:** 2024-04-17

**Authors:** Xiaxia Du, Yinuo Wang, Felix Kwame Amevor, Zifan Ning, Xun Deng, Youhao Wu, Shuo Wei, Xueqing Cao, Dan Xu, Yaofu Tian, Lin Ye, Gang Shu, Xiaoling Zhao

**Affiliations:** 1State Key Laboratory of Swine and Poultry Breeding Industry, College of Animal Science and Technology, Sichuan Agricultural University, Chengdu 611130, China; 15680811065@163.com (X.D.); wyn18128220797@163.com (Y.W.); amevorfelix@gmail.com (F.K.A.); 2020202023@stu.sicau.edu.cn (Z.N.); 2020202022@stu.sicau.edu.cn (X.D.); 15923721065@163.com (Y.W.); 13541399198@163.com (S.W.); 13835890389@163.com (X.C.); 13086605395@163.com (D.X.); maikurakitt@163.com (Y.T.); 71129@sicau.edu.cn (L.Y.); 2Farm Animal Genetic Resources Exploration and Innovation Key Laboratory of Sichuan Province, College of Animal Science and Technology, Sichuan Agricultural University, Chengdu 611130, China; 3Key Laboratory of Livestock and Poultry Multi-Omics, Ministry of Agriculture and Rural Affairs, Sichuan Agricultural University, Chengdu 611130, China; 4Department of Basic Veterinary Medicine, Sichuan Agricultural University, Chengdu 611130, China; dyysg2005@sicau.edu.cn

**Keywords:** laying hen, fatty liver, adipose tissue, lipid metabolism, inflammation

## Abstract

**Simple Summary:**

Chicken adipose tissue is essential in maintaining energy homeostasis and is also associated with the development of fatty liver. The liver and adipose tissue cooperatively promote energy balance, metabolic stability, and hormonal regulation in the body. This study investigated the effects of a high-energy low-protein (HELP) diet on laying hens as well as explored the dynamic changes in metabolism and inflammation in the liver and abdominal adipose tissue (AAT). The results showed that lipid peroxidation and inflammatory responses occur simultaneously in the liver and AATs of laying hens fed a HELP diet. This suggests a potential reciprocal regulation between the liver and AAT during the development of fatty liver in laying hens.

**Abstract:**

The aim of this study was to evaluate the effects of a high-energy low-protein (HELP) diet on lipid metabolism and inflammation in the liver and abdominal adipose tissue (AAT) of laying hens. A total of 200 Roman laying hens (120 days old) were randomly divided into two experimental groups: negative control group (NC group) and HELP group, with 100 hens per group. The chickens in the NC group were fed with a basic diet, whereas those in the HELP group were given a HELP diet. Blood, liver, and AAT samples were collected from 20 chickens per group at each experimental time point (30, 60, and 90 d). The morphological and histological changes in the liver and AAT were observed, and the level of serum biochemical indicators and the relative expression abundance of key related genes were determined. The results showed that on day 90, the chickens in the HELP group developed hepatic steatosis and inflammation. However, the diameter of the adipocytes of AAT in the HELP group was significantly larger than that of the NC group. Furthermore, the results showed that the extension of the feeding time significantly increased the lipid contents, lipid deposition, inflammatory parameters, and peroxide levels in the HELP group compared with the NC group, whereas the antioxidant parameters decreased significantly. The mRNA expression levels of genes related to lipid synthesis such as fatty acid synthase (*FASN*), stearoyl-coA desaturase (*SCD*), fatty acid binding protein 4 (*FABP4*), and peroxisome proliferator-activated receptor gamma (*PPARγ*) increased significantly in the liver and AAT of the HELP group, whereas genes related to lipid catabolism decreased significantly in the liver. In addition, the expression of genes related to lipid transport and adipokine synthesis decreased significantly in the AAT, whereas in the HELP group, the expression levels of pro-inflammatory parameters such as tumor necrosis factor-alpha (*TNF-α*), interleukin-6 (*IL-6*), and interleukin-1 beta (*IL-1β*) increased significantly in the liver and AAT. Conversely, the expression level of the anti-inflammatory parameter interleukin-10 (*IL-10*) decreased significantly in the liver. The results indicated that the HELP diet induced lipid peroxidation and inflammation in the liver and AAT of the laying hens. Hence, these results suggest that chicken AAT may be involved in the development of fatty liver.

## 1. Introduction

Fatty liver syndrome (FLS) represents a significant nutritional metabolic challenge affecting laying hens, particularly during the peak or late stages of egg production [1]. This condition compromises the health and productivity of the birds, as well as causes substantial economic losses in the poultry industry [2]. Thus, FLS is associated with several problems such as a decline in egg production, egg quality, bone quality, and birds’ health [3]. Clinical autopsy results showed an increased thickening of the abdominal wall and subcutaneous fat, enlarged liver, yellowish-colored liver, liver hemorrhage, and the presence of a large amount of fat covering the liver and heart [3].

FLS is caused by several factors such as genetics, poor feeding management, and environmental factors. In addition, excess secretion of estrogen during sexual maturity may also cause FLS in chickens, however, it is mainly due to over-nutrition and excessive energy–protein ratio [4]. The chicken liver is the main site for fat synthesis, where 90% of fat is synthesized [5]. Excessive energy–protein ratio can accelerate the conversion of acetyl coenzyme A to fat, which leads to excessive fat synthesis in the liver [6]. On the other hand, insufficient synthesis of apolipoproteins, which is probably caused by insufficient dietary protein, reduces the transport of fat from the liver to the oocyte, hence resulting in an abnormal accumulation of fat in the liver. This phenomenon causes severe fat degeneration in the liver and ultimately the occurrence of FLS [7]. Reports indicated that HELP is a common and effective approach in modeling FLS in laying hens [8,9].

Many studies have shown that adipose tissue is closely related to the central nervous system, immune system, and the endocrine system by secreting a variety of hormone signaling regulatory molecules [10,11,12]. There is a cell dialogue between adipose tissue and organs and tissues of these systems, liver, muscle cells, etc., so it has complex endocrine and metabolic functions [13]. For example, adipose tissue secretes sex hormones such as estrogen, tumor necrosis factor-α (TNF-α), leptin, and adiponectin and about 1/3 of interleukin-6 (IL-6) [14]. In addition, the levels of macrophages in the adipose tissue are positively correlated with the size and weight of adipocytes [15]. In general, adipokines regulate the interaction between adipose tissue and other metabolic organs such as the liver [16]. Therefore, adipokines directly connect with the liver through the portal vein, thereby significantly impacting liver diseases [17].

However, the expression of the parameters related to lipid metabolism and inflammation of AAT in laying hens with FLS remains unclear. Therefore, in this experiment, we modeled FLS in Roman laying hens by feeding them with a HELP diet. Thereafter, the effects of the HELP diet on the morphology and lipid metabolism of the liver and AAT of the laying hens were compared, and the differences in the synthesis of adipokines and the expression of genes related to inflammation were also determined to investigate the metabolic and inflammatory differences in the adipose tissue of FLS laying hens induced by the HELP diet.

## 2. Materials and Methods

### 2.1. Animals and Treatments

A total of 200 Roman laying hens (120 days old, >95% egg production rate) were obtained from the Chicken Breeding Unit of Sichuan Agricultural University and were randomly divided into two experimental groups (100 hens in each group), with four replicates of 25 hens in each group. Throughout the 90-day experimental period, the temperature and relative humidity in the chicken house were maintained at 20 ± 3 °C and 65–75%, respectively. The birds were kept under a photoperiod of 16 h of light and 8 h of darkness. The diet composition and nutrient levels of the NC and HELP groups are shown in Table 1. The HELP diet contained 3.58% less protein and 422 kcal/kg more energy. Drinking water was provided ad libitum. In addition, daily egg production was recorded.

### 2.2. Sample Collection and Measurement

On the 30th, 60th, and 90th day of the experiment, 20 laying hens from each experimental group were randomly selected for sampling. Before the sample collection, the birds were fasted for 12 h and weighed. The blood samples were collected from the wing vein, and the serum was collected after centrifugation and was subsequently stored at −80 °C. Thereafter, the birds were anesthetized and subsequently euthanized. Liver tissues and AAT were carefully excised from each bird. Whole liver and standardized portions of the AAT were immediately weighed to obtain their fresh weight. For consistency and to facilitate comparative analyses, representative samples of AAT and liver weighing approximately 10 g and 20 g, respectively, were collected from each bird. Those tissue samples were temporarily frozen in liquid nitrogen and subsequently stored at −80 °C for further RNA extraction, and the other parts of these samples were cut into 1 cm^3^ pieces and fixed in 4% paraformaldehyde for subsequent tissue sectioning.

### 2.3. Morphological and Histological Observation of the Liver and AATs

The fixed liver tissues and AATs were dehydrated with an alcohol gradient and were made transparent with xylene. The transparent samples were embedded in paraffin, cut into 5 mm slices, and routinely stained with hematoxylin and eosin (H&E) and Oil Red O. All the sections were viewed under an electronic microscope (DP80Digital, Olympus, Tokyo, Japan), and then ten fields were randomly selected for statistical analysis.

### 2.4. Ultrastructural Observation of the Liver and AATs

The fresh liver tissues and AATs (at day 90) were prefixed with 3% glutaraldehyde, refixed with 1% osmium tetroxide, dehydrated in acetone, embedded in Epon812, then the semi-thin sections were optically localized, and the ultrathin sections were double stained with uranyl acetate and lead citrate and were finally observed using the Hitachi H-600V transmission electron microscope (Tokyo, Japan).

### 2.5. RNA Isolation and Quantitative Real-Time PCR (qRT-PCR)

Total RNA was isolated from all the tissue samples according to the instructions of the Trizol reagent (Takara, Tokyo, Japan). The RNA concentration and purity were estimated by determining the A260/A280 absorbance ratio and the 18 S and 28 S bands in a 1% agarose gel. Thereafter, reverse transcription was performed using a reverse transcriptase enzyme and random primers, following the manufacturer’s instructions. The reaction was carried out at specific temperature and for a specific time. Then, gene-specific primers were designed using the Primer Premier 5.0 software according to the coding sequences of the target genes (Table 2). A qRT-PCR reaction was performed using a real-time PCR machine. The reaction mixture included cDNA template, gene-specific primers, a fluorescent dye, and a reaction buffer. Cycling conditions involved an initial denaturation step, followed by a certain number of amplification cycles, with specific denaturation, annealing, and extension temperatures and times. Fluorescent signals were recorded during each cycle. Data analysis was conducted by determining the Ct values (cycle threshold) for each sample, representing the cycle number at which the fluorescent signal crossed a defined threshold. The expression level of the target gene was normalized to the expression level of reference genes, selecting appropriate reference genes. Relative gene expression levels were calculated using the 2^−∆∆Ct^ method [18]. Statistical analysis was performed to determine significant differences between the experimental groups.

### 2.6. Determination of Biochemical Parameters

The levels of the total cholesterol (TC), triglyceride (TG), very low-density lipoprotein y (VLDLy), high-density lipoprotein cholesterol (HDL-ch), alanine aminotransferase (ALT), aspartate aminotransferase (AST), malondialdehyde (MDA), lipid peroxidation (LPO), total superoxide dismutase (T-SOD), glutathione (GSH), tumor necrosis factor-α (TNF-α), and interleukin-6 (IL-6) were determined in the serum and the liver tissue using enzyme-linked immunosorbent assay (ELISA) kits (Andygene, Beijing, China), following the manufacturer’s instruction.

### 2.7. Statistical Analysis

All data were analyzed using SPSS 19.0 Statistics Software (SPSS, Chicago, IL, USA). The results were represented as the mean ± standard error of the mean (SEM). Each treatment had at least three replicates. The significance between the two groups was determined using an unpaired Student’s *t*-test. For multiple groups, a one-way ANOVA method was used, and the significance level was set at * *p* < 0.05, ** *p* < 0.01.

## 3. Results

### 3.1. Production Performance and Organ Weights

The results obtained from this study showed that the egg production of the chickens decreased significantly on the 10th to 90th day of the experiment in the HELP group as compared to the NC group (Figure 1A). On the 90th day of the experiment, the results showed that the laying rate of chickens in the NC group was 99%, whereas the laying rate of the chickens in the HELP group was 73%. Further, the results showed that the body weights, liver weights, and AAT weights of laying hens in the HELP group were increased significantly on the 90th day of feeding compared to the NC group (Figure 1B–D).

### 3.2. Morphological and Histological Observations of the Chicken Liver and AAT 

Morphological observations revealed that the livers of the laying hens in the NC group showed a reddish brown color and less intra-abdominal fat deposition, whereas the livers of the HELP group increased in size and appeared to be earthy yellow in color, while a large amount of intra-abdominal adipose tissue was deposited with the prolongation of the feeding time of HELP (Figure 2A). In the NC group, the histological analysis showed that the structure of the liver tissue was normal, the hepatocytes were arranged neatly and clearly, and the cytoplasm of the hepatocytes was pink and evenly distributed (Figure 2B). On the 90th day of the experiment, the hepatocytes of the HELP group were enlarged, the cytoplasm was loose, and there were a large number of fat vacuoles (Figure 2B). The Oil Red O staining results also showed that with the extension of the high-fat feeding time, a large number of lipid droplets were deposited in the hepatocytes of the chickens in the HELP group (Figure 2C), indicating that the livers of the chickens in the HELP group were steatotic. Moreover, the histological results of the AAT showed that the diameter of the adipocytes gradually increased with the extension of the feeding time of the HELP diet, and the adipocytes in the HELP group were significantly larger than those in the NC group at d90 (Figure 2D,E).

### 3.3. Ultrastructural Observation of the Chicken Liver and AAT

The phenotypic results of the liver showed the presence of steatosis, and the diameter of the adipocytes was increased significantly on the 90th day. Thereafter, tissue ultrastructures of the liver and AATs were observed on the 90th day, and the results from the electron microscopy showed that there were a large number of lipid droplets with different sizes in the hepatocytes of the chickens in the HELP group compared with the NC group (Figure 3A). The results also showed that the nuclei of the adipocytes in the HELP group were extruded and deformed due to excessive enlargement of the lipid droplets, and the mitochondria were observed in the cells. However, the nuclei of the adipocytes in the AAT of the NC group were ovals, and the mitochondria were observed in the cytoplasm (Figure 3B). These results indicated that HELP diet caused significant changes in the ultrastructure of the chicken AAT, as well as decreased its metabolic activity.

### 3.4. Determination of the Serum Biochemical Parameters

On the 30th, 60th, and 90th day of the experiment, the levels of serum TC and TG of the birds in the HELP group were significantly higher than those in the NC group (*p* < 0.01, Figure 4A,B). Compared with the NC group, the levels of serum ALT and AST increased significantly in the HELP group (*p* < 0.05, Figure 4C,D). In addition, the levels of T-SDO and GSH in the serum decreased significantly in the HELP group as compared with the NC group (*p* < 0.05, Figure 4E,F), however, the levels of serum TNF-α and IL-6 increased significantly in the HELP group (*p* < 0.05, Figure 4G,H).

### 3.5. Determination of Liver Biochemical Parameters

The biochemical parameters such as TC, TG, VLDLy, HDL-ch, MDA, LPO, T-SOD, and GSH were measured in the liver using ELISA kits (Figure 5). The results showed that the levels of TC, TG, and VLDLy were significantly higher in the livers of chickens in the HELP group at 30, 60, and 90 days of the experiment (*p* < 0.01, Figure 5A–C). However, the level of HDL-ch was significantly reduced (*p* < 0.01, Figure 5D). Cellular damage and mitochondrial dysfunction caused by hepatocyte lipid metabolism disorder can progressively up-regulate the level of oxidative stress in the liver, hence, the results obtained in this present study showed that the level of MDA and LPO increased significantly in the liver of chickens in the HELP group at d90 as compared with those in the NC group (*p* < 0.01, Figure 5E,F). However, at d90, the levels of T-SOD and GSH decreased significantly in the liver of chickens in the HELP group as compared with the NC group (*p* < 0.01, Figure 5G,H).

### 3.6. Expression Abundance of Genes Related to Lipid Metabolism in the Liver

The expression of genes related to lipid metabolism in the chicken liver was determined on d30, 60, and 90, and the results showed that the expression abundance of genes related to lipolysis such as *PPARα*, *MTTP*, and *SERBP1* decreased significantly in the HELP group as compared to that in the NC group, however, the expression abundance of genes related to lipid synthesis such as *PPARγ*, *FASN*, and *SCD* increased significantly in the HELP group compared to the NC group (*p* < 0.01, Figure 6A–F). Furthermore, the mRNA expression of genes related to lipid transport including *VTGII*, *ApoB*, and *LDLR* decreased significantly in the liver of the birds in the HELP group as compared to those in the NC group on d90 (*p* < 0.01, Figure 6G–I).

### 3.7. Expression Abundance of Genes Related to Lipid Metabolism and Adipokine Synthesis in the AAT 

Figure 7 shows the expression of genes related to lipid metabolism and adipokine synthesis in the AAT on d30, 60, and 90 of the experiment, and the results showed that the mRNA levels of genes related to lipid synthesis (*FABP4* and *PPARγ*) increased significantly (*p* < 0.01, Figure 7A,D) in the HELP group as compared with the NC group, whereas the mRNA levels of genes related to lipolysis (*LPL* and *CPT-1A*) decreased significantly in the AAT of birds in the HELP group throughout the various experimental time points (*p* < 0.01, Figure 7B,E). In addition, the expression of genes related to fat synthesis (*FABP4* and *PPARγ*) increased significantly, whereas the expression of genes related to adipokine synthesis such as *LEPTIN* and *ADPN* decreased significantly (*p* < 0.01, Figure 7C,F).

### 3.8. Expression Abundance of Inflammation-Related Genes in the Liver and AAT

As shown in Figure 8 and Figure 9, the mRNA levels of pro-inflammatory-related genes such as tumor necrosis factor-alpha (*TNF-α*), interleukin-1β (*IL-1β*), and interleukin-6 (*IL-6*) increased significantly (*p* < 0.05) in the liver and AAT of the birds in the HELP group, however, the expression of anti-inflammatory-related genes such as interleukin-10 (*IL-10*) decreased significantly in the liver of the birds in the HELP group compared to that in the NC group (*p* < 0.05).

## 4. Discussion

FLS is mainly attributed to lipid metabolism disorder in laying hens. The liver is the main organ that synthesizes lipids as precursors for egg yolk formation; however, excessive deposition and accumulation of lipids may impair liver function, leading to lipid metabolism disorders [19]. Excessive feeding of a HELP diet has been demonstrated to contribute to FLS in hens [20]. In this study, Roman laying hens were fed with a HELP diet for 90 days, which significantly induced hepatic lipidosis. Further, results from the histological observation confirmed the accumulation of excessive fat vacuoles and lipid droplets in the liver, as well as abnormal liver function markers, indicating successful modeling of the FLS.

Adipose tissue plays a crucial role in energy metabolism and storage. Excessive energy intake, as observed in our study, leads to abnormal adipocyte metabolism and enhanced triglyceride accumulation in both the liver and adipose tissue. This is consistent with previous findings that link excessive energy intake to FLS development in laying hens [20]. Our results highlight a potential reciprocal regulation between the liver and AAT during the progression of FLS, suggesting that interventions aimed at modulating adipose tissue metabolism could mitigate FLS severity.

Adipose tissue is not only the major site for energy storage but also an active endocrine organ, secreting various adipokines that regulate metabolic homeostasis [21,22,23,24]. In the context of FLS, excessive lipid accumulation disrupts adipose tissue homeostasis, leading to altered adipokine profiles and contributing to liver steatosis [25,26]. Our present study extends these findings by demonstrating that a HELP diet significantly alters the expression of genes related to adipokine synthesis in the AAT, supporting the hypothesis that adipose tissue dysfunction contributes to the pathogenesis of FLS in laying hens. Furthermore, our investigation into the expression of inflammation-related genes in the liver and AAT reveals a pronounced pro-inflammatory state in the hens fed the HELP diet. This inflammation may exacerbate lipid accumulation and adipose tissue dysfunction, further contributing to FLS progression. Our findings suggest that targeting inflammatory pathways could offer therapeutic potential in managing FLS in laying hens.

Mitochondrial dysfunction has emerged as a key factor in the pathogenesis of NAFLD and FLS, with impaired fatty acid oxidation and oxidative stress playing pivotal roles [27,28,29]. This present study corroborates these observations, showing significant mitochondrial alterations and oxidative stress in hens subjected to the HELP diet. These results underscore the importance of maintaining mitochondrial function and managing oxidative stress as strategies to prevent FLS in laying hens.

In conclusion, the present study provides compelling evidence of the interconnected roles of lipid peroxidation, inflammation, and mitochondrial dysfunction in the development of FLS in laying hens fed a HELP diet. These findings underscore the complexity of FLS pathogenesis and highlight potential targets for intervention. Future studies should explore therapeutic strategies aimed at restoring adipose tissue and liver homeostasis to potentially mitigates the impacts of FLS in the poultry industry.

## 5. Conclusions

Taken together, lipid peroxidation and inflammatory responses occur simultaneously in the liver and AATs of the laying hens fed with the HELP diet, suggesting a potential reciprocal regulation between the liver and AAT during fatty liver development in laying hens.

## Figures and Tables

**Figure 1 animals-14-01199-f001:**
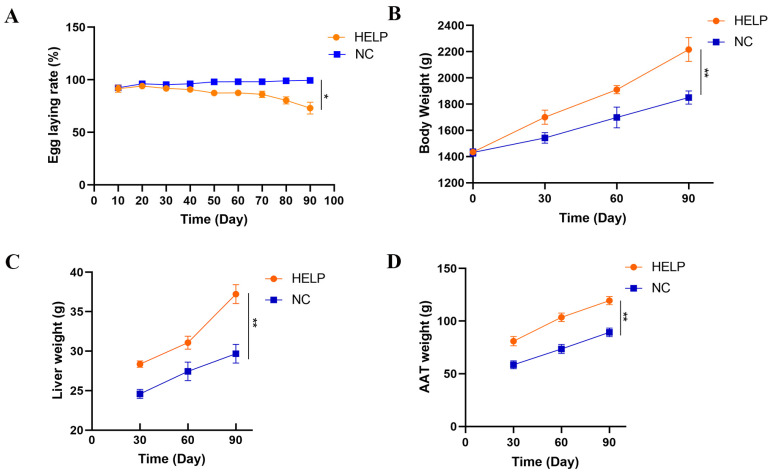
Performance and organ weight of the laying hens fed basal and HELP diets for 30, 60, and 90 d. (**A**) Egg-laying rate. (**B**) Body weight. (**C**) Liver weight. (**D**) AAT weight. Data are expressed as the mean ± SE (n = 20, each group). * *p* < 0.05, ** *p* < 0.01.

**Figure 2 animals-14-01199-f002:**
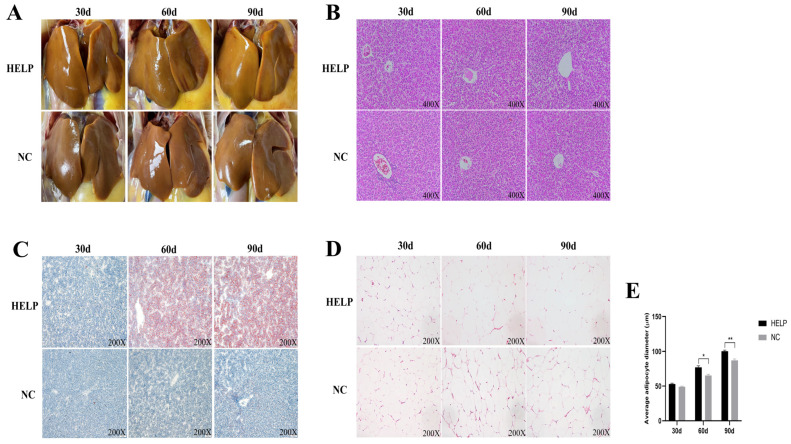
Morphological changes in the AAT and liver of chickens fed basal and HELP diets for 30, 60, and 90 days. (**A**) Morphological changes in the chicken liver and AAT. (**B**) Histological changes in the chicken liver (magnified 400×). (**C**) Oil Red O staining of frozen slices of the chicken liver (magnified 200×). (**D**) Histological changes in the adipose tissue of the chicken abdomen (magnified 200×). (**E**) Comparative analysis of the lipid droplet diameter in the AAT of chickens. Data are expressed as the mean ± SE (n = 6, each group). * *p* < 0.05, ** *p* < 0.01.

**Figure 3 animals-14-01199-f003:**
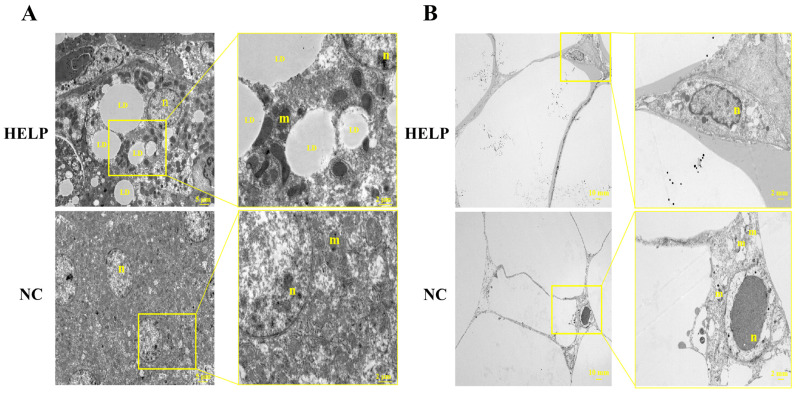
Transmission electron microscopy of the AAT and liver of the chickens fed with normal and HELP diets for a period of 90 days. (**A**) Transmission electron microscopy of the chicken liver. (**B**) Transmission electron microscopy of AAT in chickens. Cell nucleus: n, mitochondria: m, lipid droplets: LD.

**Figure 4 animals-14-01199-f004:**
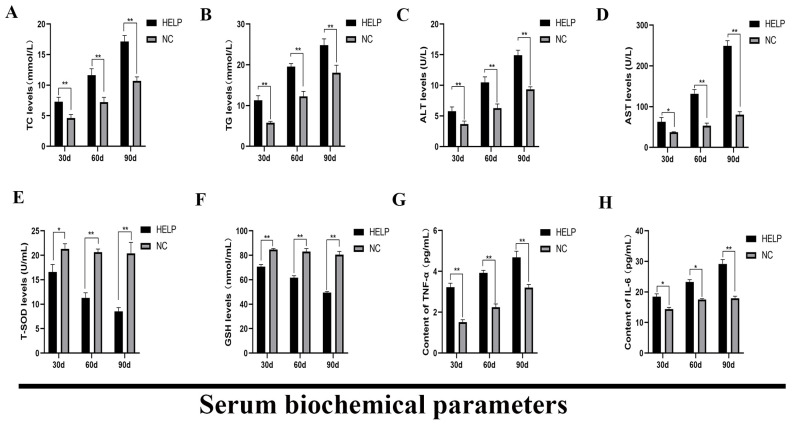
Effects of HELP diet on the levels of serum biochemical parameters of laying hens. (**A**) Total cholesterol, TC. (**B**) Triglyceride, TG. (**C**) Alanine aminotransferase, ALT. (**D**) Aspartate aminotransferase, AST. (**E**) Total superoxide dismutase, T-SOD. (**F**) Glutathione, GSH. (**G**) Tumor necrosis factor-α, TNF-α. (**H**) Interleukin-6, IL-6. Data are expressed as the mean ± SE (n = 9, each group). * *p* < 0.05, ** *p* < 0.01.

**Figure 5 animals-14-01199-f005:**
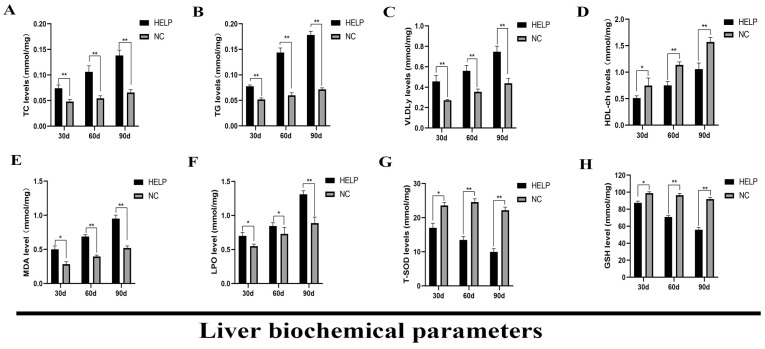
Impacts of feeding HELP diet on the liver biochemical parameters of laying hens. (**A**) Total cholesterol, TC. (**B**) Triglyceride, TG. (**C**) Very low-density lipoprotein y, VLDLy. (**D**) High-density lipoprotein cholesterol, HDL-ch. (**E**) Malondialdehyde, MDA. (**F**) Lipid peroxide, LPO. (**G**) Total superoxide dismutase, T-SOD. (**H**) Glutathione, GSH. Data are expressed as the mean ± SE (n = 9, each group). * *p* < 0.05, ** *p* < 0.01.

**Figure 6 animals-14-01199-f006:**
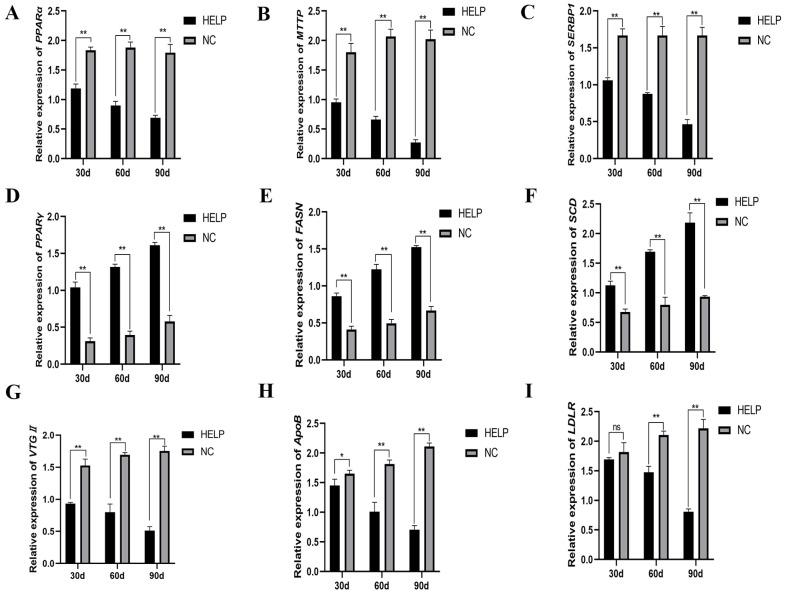
The relative expression of genes related to lipolysis, lipid synthesis, and lipid transport in the liver of chickens fed with or without HELP diet. (**A**–**C**) The mRNA expression of genes related to lipolysis (*PPARα*, *MTTP*, and *SERBP1*). (**D**–**F**) The mRNA expression of genes related to lipid synthesis (*PPARγ*, *FASN*, and *SCD*). (**G**–**I**) The expression of genes related to lipid transport (*VTGII*, *ApoB*, and *LDLR*). Data are expressed as the mean ± SE (n = 9, each group). * *p* < 0.05, ** *p* < 0.01. Differences marked with ‘ns’ are not statistically significant.

**Figure 7 animals-14-01199-f007:**
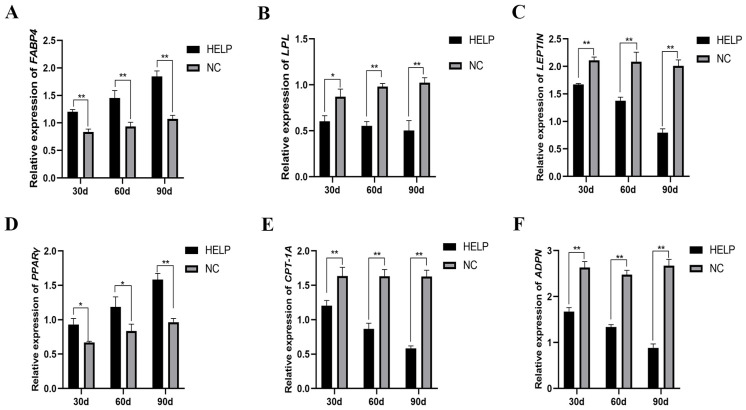
The mRNA expression of genes related to lipid synthesis and hormone synthesis in the AAT of chickens fed with or without HELP diet. The mRNA expression of *FABP4* (**A**), *LPL* (**B**), *LEPTIN* (**C**), *PPARγ* (**D**), *CPT-1A* (**E**), and *ADPN* (**F**) in the AAT. Data are expressed as the mean ± SE (n = 9, each group). * *p* < 0.05, ** *p* < 0.01.

**Figure 8 animals-14-01199-f008:**
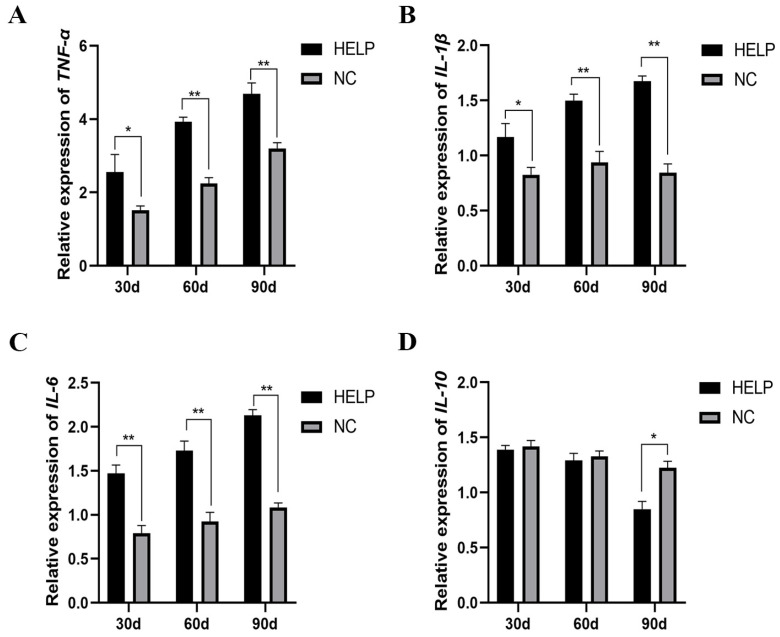
The mRNA expression of genes related to inflammatory parameters in the liver of chickens fed with or without HELP diet. The mRNA expression of *TNF-α* (**A**), *IL-1β* (**B**), *IL-6* (**C**), *IL-10* (**D**) in the AAT. Data are expressed as the mean ± SE (n = 9, each group). * *p* < 0.05, ** *p* < 0.01.

**Figure 9 animals-14-01199-f009:**
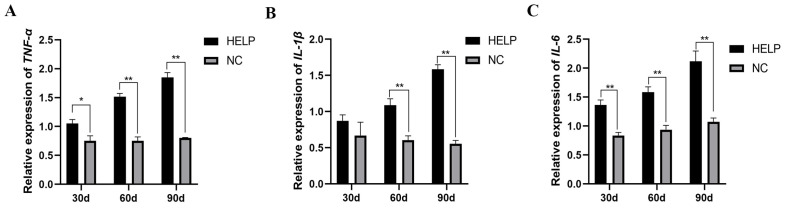
The mRNA expression of genes related to inflammatory parameters in the AAT of chickens fed with or without HELP diet. The mRNA expression of *TNF-α* (**A**), *IL-1β* (**B**), *IL-6* (**C**) in the AAT. Data are expressed as the mean ± SE (n = 9, each group). * *p* < 0.05, ** *p* < 0.01.

**Table 1 animals-14-01199-t001:** Composition and nutritional levels of the diets (dry matter basis) %.

Items %	NC Group	HELP Group
Corn (maize)	64.00	70.00
Wheat bran	2.00	1.20
Soybean meal	24.00	14.58
Soybean oil	0	5
Calcium	8.00	8.00
* Premix	2.00	2.00
Total	100.00	100.00
Nutrient level		
Crude protein (CP)	15.58	12.00
Available phosphate (AP)	0.51	0.46
Arginine (Arg)	1.03	0.74
Methionine (Met)	0.37	0.32
Valine (Val)	0.77	0.58
Metabolic energy (kcal/kg)	2678.99	3100.00
Methionine + cysteine (Met + Cys)	0.67	0.56

* Each kilogram of feed contains the following elements: copper, 2.50 mg; iron, 20.00 mg; zinc, 17.50 mg; manganese, 15.00 mg; potassium iodide, 4.00 mg; sodium selenite, 6.00 mg; methionine, 50.00 mg; pyrimidine, 2.00 mg; multidimensional, 15.00 mg; phytase, 10.00 mg; kallikrein, 7.50 mg; antioxidant, 2.00 mg; choline, 50.00 mg; salt, 200.00 mg; calcium phosphate, 500.00 mg; zeolite powder, 76.00 mg.

**Table 2 animals-14-01199-t002:** Primers used for the quantitative real-time PCR (qRT-PCR).

Gene	Sequence (5′–3′)	Product Length (bp)	Annealing Temperature (°C)	Accession Number
*PPARα*	F: AGGCCAAGTTGAAAGCAGAA R: TTTCCCTGCAAGGATGACTC	155	60	NM_001001464.1
*PPARγ*	F: TGACAGGAAAGACGACAGACA R: CTCCACAGAGCGAAACTGAC	164	59	NM_001001460.1
*MTTP*	F: GTTCTGAAGGACATGCGTGC R: GATGTCTAGGCCGTACGTGG	120	58	NM_001109784.2
*SREBP1*	F: CTACCGCTCATCCATCAACG R: CTGCTTCAGCTTCTGGTTGC	136	60	NM_204126.3
*FASN*	F: TGCTATGCTTGCCAACAGGA R: ACTGTCCGTGACGAATTGCT	128	59	NM_205155.3
*SCD*	F: CTATGCGGGGCTACTT R: GGATGGCTGGAATGAA	167	58	NM_204890.2
*VTGII*	F: AACTACTCGATGCCCGCAAA R: ACCAGCAGTTTCACCTGTCC	179	58	NM_001031276.1
*ApoB*	F: GGTTACTCCCACGATGGCAA R: TCGCAGAAATGCCCTTCCTT	120	60	NM_001044633.2
*LDLR*	F: GTGGACGAGTGCTCTCAGG R: ATAGAGGTTCCCTTCGGCCA	167	58	NM_204452.1
*FABP4*	F: GCCTGACAAAATGTGCGACC R: ATTAGGCTTGGCCACACCAG	130	59	NM_204290.2
*LPL*	F: GCATTCACCATTCAGAGAGTCAG R: AACTGCTAAAGAGGAACTGATGG	130	57	NM_205282.2
*CPT-1A*	F: TGAGCACTCTTGGGCAGATG R: TCTCCTTTGCAGTGTCCGTC	108	56	NM_001012898.1
*LEPTIN*	F: GCAGTGCCGTGCCA GACT C R: GA ATGTCCTGCAGAGAGCCC	284	58	LN794246.1
*ADPN*	F: CCCAGAGCAGTGGCTGTTTA R: TGGGTATTTCCAAGGGACGC	116	59	NM_206991.2
*TNFα*	F: TGTGCTGTGTGCAACGACTA R: CAGGCCTGGCAACTCTTTCT	144	60	NM_205183.2
*IL-1β*	F: TGCCTGCAGAAGAAGCCTCG R: GACGGGCTCAAAAACCTCCT	205	56	NM_204524.1
*IL-6*	F: CTGCAGGACGAGATGTGCAA R: AGGTCTGAAAGGCGAACAGG	166	57	NM_204628.1
*IL-10*	F: AGTTTAAGGGGACCTTTGGCT R: AACTCCCCCATGGCTTTGTAG	265	59	XM_025143715.1
*β-actin*	F: TATTGCTGCGCTCGTTGTTG R: TGGCCCATACCAACCATCAC	144	60	NM_205518.1

*PPARα/γ:* peroxisome proliferator-activated receptor alpha/gamma; *MTTP:* microsomal triglyceride transport protein; *SREBP1:* sterol regulatory element binding proteins 1c; *FASN*: fatty acid synthase; *SCD*: acyl-coA desaturase; *VTGII*: vitellogenin-II; *ApoB*: apoliprotein B; *LDLR*: low-density lipoprotein receptor; *FABP4*: fatty-acid-binding protein 4; *LPL*: lipoprotein lipase; *CPT-1A*: carnitine palmitoyl transferase 1A; *ADPN*: adiponectin; *TNFα*: tumor necrosis factor-α; *IL-1β/6/10*: interleukin-1β/6/10; F: forward primer; R: reverse primer.

## Data Availability

Data are contained within the article.

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
