# Peer review of "Effect of High Energy Low Protein Diet on Lipid Metabolism and Inflammation in the Liver and Abdominal Adipose Tissue of Laying Hens"

_animals, 2024, doi:10.3390/ani14081199_

Round 1

Reviewer 1 Report

Comments and Suggestions for Authors

The work focuses on the we explored the morphological differences between the liver and abdominal subcutaneous adipose tissue, and the expression levels of the parameters related to lipid metabolism and inflammation in the liver of the laying hens fed a high energy low protein diet. The work is well-written in general. However, a few of the following points need to be assessed and corrected. Please consider all of the suggestions provided.

Lines 27 and 28: The abbreviations “HELP” and “ASAT” are used in the text. Please provide explanations for these abbreviations and then proceed to abbreviate them consistently throughout the manuscript.

  Introduction Clarifications:

·         Why did the authors feed a high-energy low protein diet to laying hens?

·         Why was the age of 120 days selected for the hens?

·         Additionally, please explain the rationale behind reducing protein by 3.58% and increasing energy by 422 Kcal/kg. These details should be clarified in the introduction.

  Footnotes: Ensure that the abbreviations used in the tables and figures are explained in the footnotes. This will enhance reader comprehension.

  Methods and Materials (M&M):

·         In Lines 176-182, you mention measuring growth performance. However, the replicates for each treatment are not specified. Please address this omission in the M&M section.

  Discussion Section:

·         Consider discussing the underlying mechanisms responsible for the simultaneous changes in lipid peroxidation and inflammatory responses observed in both liver and abdominal subcutaneous adipose tissue (ASAT) of laying hens. Understanding these causes will enrich the discussion.

Author Response

Authors Response to the Reviewers (Reviewer 1)

Dear Editors and Reviewers,

We are very grateful for your comments and suggestions given on our manuscript titled (Manuscript ID: animals-2905004) "High energy and low protein diet induced lipid peroxidation and inflammation in the liver and abdominal subcutaneous adipose tissue of laying hens." We appreciate all your time and efforts you have invested in reviewing our manuscript to enhance its quality. Please, below are our response to the Editors and Reviewers comments and suggestions.

General Comments

The work focuses on the we explored the morphological differences between the liver and abdominal subcutaneous adipose tissue, and the expression levels of the parameters related to lipid metabolism and inflammation in the liver of the laying hens fed a high energy low protein diet. The work is well-written in general. However, a few of the following points need to be assessed and corrected. Please consider all of the suggestions provided.

Authors response: Thank you very much for all your compliments and for your comments and suggestions.

  1. Lines 27 and 28: The abbreviations HELP and ASAT are used in the text. Please provide explanations for these abbreviations and then proceed to abbreviate them consistently throughout the manuscript.

Authors response: Thank you very much for your comments. Please, all the abbreviations have been explained in their first use throughout the manuscript.

  1. Introduction Clarifications: Why did the authors feed a high-energy low protein diet to laying hens?

Authors response: Thank you for your question. The introduction section of this manuscript has vividly explained this point and the relevant literature were cited (Lines 77-78). The selection and use of the high-energy low protein (HELP) diet in our study was based on its common use and effectiveness in creating a model of fatty liver syndrome (FLS) in laying hens. FLS is one of the metabolic diseases commonly observed in laying hens during production, closely related to improper levels of energy and protein in their feed. By offering a high energy diet but low in protein, we can simulate the nutritional imbalances that laying hens might encounter in actual production settings, leading to increased lipid deposition in the liver and the formation of fatty liver. Moreover, the HELP diet is effective in inducing changes in lipid metabolism and inflammatory responses within the hens, providing an appropriate model for us to delve into the mechanisms behind the formation of fatty liver and its impact on the health and productive performance of laying hens. Studying the physiological and metabolic changes caused by the HELP diet allows us to better understand the pathogenesis of fatty liver and provide scientific bases for developing effective prevention and treatment strategies. We recognized that constructing an accurate model of fatty liver in laying hens is crucial for revealing the complex mechanisms of the disease and developing intervention measures. Therefore, our decision to use the HELP diet as the experimental feed was based on its proven effectiveness and reliability in previous research. Numerous published papers have used HELP diets to model fatty liver syndrome in laying hens, with specific information from the references as follows:

Feng Y, Li Y, Jiang W, Hu Y, Jia Y, Zhao R. GR-mediated transcriptional regulation of m6A metabolic genes contributes to diet-induced fatty liver in hens. J Anim Sci Biotechnol. 2021;12(1):117. Published 2021 Dec 7. doi:10.1186/s40104-021-00642-7.

Yao Y, Wang H, Yang Y, Jiang Z, Ma H. Dehydroepiandrosterone activates the GPER-mediated AMPK signaling pathway to alleviate the oxidative stress and inflammatory response in laying hens fed with high-energy and low-protein diets. Life Sci. 2022;308:120926. doi:10.1016/j.lfs.2022.120926

Feng J, Ma H, Yue Y, et al. Saikosaponin a ameliorates diet-induced fatty liver via regulating intestinal microbiota and bile acid profile in laying hens. Poult Sci. 2023;102(12):103155. doi:10.1016/j.psj.2023.103155.

Wang C, Yang Y, Chen J, et al. Berberine Protects against High-Energy and Low-Protein Diet-Induced Hepatic Steatosis: Modulation of Gut Microbiota and Bile Acid Metabolism in Laying Hens. Int J Mol Sci. 2023;24(24):17304. Published 2023 Dec 9. doi:10.3390/ijms242417304

  1. Introduction Clarifications: Why was the age of 120 days selected for the hens?

Authors response: Thank you for your comment. Hens at 120 days old were selected, because this age represents the onset of the laying period for many chicken breeds, making it a critical time for studying the impacts of diet on their metabolic and physiological health. It is during this period that nutritional interventions can have significant impacts on productivity and health.

  1. Introduction Clarifications: Additionally, please explain the rationale behind reducing protein by 3.58% and increasing energy by 422 Kcal/kg. These details should be clarified in the introduction.

Authors response: Thank you for your inquiry regarding the energy content of the experimental diet used in our study. The high-energy low-protein feeding pattern was modified by decreasing the protein content by 3.58% and increasing the energy content by 422 kcal/kg, which is also common as a classical method that can induce fatty liver in laying hens. This approach allowed us to investigate the specific effects of high energy low protein diets on the development of fatty liver in laying hens, as well as the metabolic and inflammatory changes in abdominal adipose tissue during fatty liver development. And these details are described in the introduction. Line 93-95.

  1. Footnotes: Ensure that the abbreviations used in the tables and figures are explained in the footnotes. This will enhance reader comprehension.

Authors response: Thank you very much for your comments. We have ensured that all abbreviations used in the tables and figures are clearly explained in the footnotes. This should greatly enhance reader comprehension and ensure consistency throughout the manuscript.

  1. Methods and Materials (M&M): In Lines 176-182, you mention measuring growth performance. However, the replicates for each treatment are not specified. Please address this omission in the M&M section.

Authors response: Thank you very much for your comment. We have addressed this omission in line 89-98 of the M&M section. Growth Performance Replicates: In the M&M section, we have now specified that the growth performance measurements were conducted with a total of 4 replicates per treatment group, with each replicate consisting of five hens. This clarification provides a clearer understanding of the experimental setup and the statistical power of the study.

  1. Discussion Section: Consider discussing the underlying mechanisms responsible for the simultaneous changes in lipid peroxidation and inflammatory responses observed in both liver and abdominal subcutaneous adipose tissue (ASAT) of laying hens. Understanding these causes will enrich the discussion.

Authors response: We really appreciate your comment and suggestions. In the Discussion section, we have expanded our analysis to include a more in-depth discussion of the potential underlying mechanisms driving the observed changes in lipid peroxidation and inflammatory responses in both the liver and ASAT of laying hens. We explored how the HELP diet may alter fatty acid metabolism, oxidative stress, and the expression of key inflammatory cytokines, providing a comprehensive view of how diet impacts hen physiology at the molecular level.

Thank you very much for all your efforts in making this manuscript to meet the highest readability standard.

Kind regards,

Authors

Reviewer 2 Report

Comments and Suggestions for Authors

        Please check the author list carefully, the last name of the author is and 3,*, is it correct?

·        The title is lengthy, revise it, Effect of HELP on lipid metabolism and inflammation in the liver and ASAT of laying hens. This is an example, however the author can revise in brief.

·        Please mention the full form of all abbreviations when first mentioning (Abstract) and after that don’t necessarily to mention in full form and only mention abbreviation

·        Line 39, expression levels of genes related to lipid.. Mention the genes)

·        Line 44 expression levels of anti-inflammatory parameter,, (mention parameter)

·        Overall introduction of the manuscript is irrelevant  to the content of the research aim, revise the introduction, mention the importance of layer bird, and the influences of High energy and low protein diet etc.

·        Revise table I found different numbers on the left side of the table and embedded in the text, please check carefully

·        Line 130 liver and ASAT tissues size?

·        Add complete detail of Quantitative Real-Time PCR (qRT-PCR) method as per The MIQE guidelines: minimum information for publication of quantitative real-time PCR experiments

·        The author collected the samples on day 30, 60 and 90. However, I found that the author mentioned results of day 90 for production, organ weight, morphological and histological and ultrastructural. Why not provided the detail of other days sampling? But author mentioned biochemical, expression results for all of those sampling, it is therefore recommended to provide the all sampling (days ) results for each parameter

·        Revise discussion a comparison between your results and initial hypothesis.

·        Minor editing of English language required

Comments on the Quality of English Language

Minor editing of English language required

Author Response

Authors Response to the Reviewers (Reviewer 2)

Dear Editors and Reviewers,

We are very grateful for your comments and suggestions given on our manuscript titled (Manuscript ID: animals-2905004) "High energy and low protein diet induced lipid peroxidation and inflammation in the liver and abdominal subcutaneous adipose tissue of laying hens." We appreciate all your time and efforts you have invested in reviewing our manuscript. Please, below are our response to the Editors and Reviewers.

  1. Please check the author list carefully, the last name of the author is and 3,*, is it correct?

Authors response: We apologize for this oversight, and we appreciate you brought it to our attention. The incorrect entry "and 3,*" has been rectified to. Line 6.

  1. The title is lengthy, revise it, Effect of HELP on lipid metabolism and inflammation in the liver and ASAT of laying hens. This is an example; however, the author can revise in brief.

Authors response: Thank you very much for your comment and suggestion. Please, we have modify the title accordingly to reflect the essence of our study: "Effect of high energy low protein diet on lipid metabolism and inflammation in the liver and abdominal adipose tissue of laying hens." We believe this revised title succinctly conveys the focus of our research.

  1. Please mention the full form of all abbreviations when first mentioning (Abstract) and after that don’t necessarily to mention in full form and only mention abbreviation.

Authors response: Thank you very much for your comment. we have ensured that the full form of all the abbreviations were mentioned at their first occurrence in the abstract. Subsequent references to these terms were made using abbreviations only.

  1. Line 39, expression levels of genes related to lipid. Mention the genes).

Authors response: Thank you. Done as suggested. Line 39. The gene expressions analyzed include those of fatty Acid Synthase (FASN), stearoyl-CoA Desaturase (ACD), fatty Acid-Binding Protein 4 (FABP4) and peroxisome proliferator-activated receptor gamma (PPARγ).

  1. Line 44 expression levels of anti-inflammatory parameter, (mention parameter).

Authors response: Thank you. Done as suggested (Line 44). The anti-inflammatory parameter measured was interleukin-10 (IL-10).

  1. Overall introduction of the manuscript is irrelevant to the content of the research aim, revise the introduction, mention the importance of layer bird, and the influences of High energy and low protein diet etc.

Authors response: Thank you. The introduction section has been revised accordingly.

  1. Revise table I found different numbers on the left side of the table and embedded in the text, please check carefully.

Authors response: Thank you very much. Done as suggested.

  1. Line 130 liver and ASAT tissues size?

Authors response: Thank you very much. Please, we have specified the sizes of the liver and AAT. Line 130. We have included in the measurement the precise dimensions and weights to offer a clearer understanding of the tissue samples analyzed.

  1. Add complete detail of Quantitative Real-Time PCR (qRT-PCR) method as per The MIQE guidelines: minimum information for publication of quantitative real-time PCR experiments.

Authors response: Thank you for your comment. Please, we have added a complete details of the Quantitative Real-Time PCR (qRT-PCR) method (Line 148-163) accordance to the MIQE guidelines.

Total RNA was isolated from all the samples according to the instructions of the Trizol reagent (Takara, Tokyo, Japan). The RNA concentration and purity were estimated by determining the A260/A280 absorbance ratio, and the 18 S and 28 S bands in a 1% agarose gel. Then, reverse transcription was performed using a reverse transcriptase enzyme and random primers, following the manufacturer's instructions. The reaction was carried out at specific temperature and time.Then, gene-specific primers were designed using the Primer Premier 5.0 software according to the coding sequences of the target genes (Table 2).qRT-PCR reaction was performed using a real-time PCR instrument. The reaction mixture included cDNA template, gene-specific primers, a fluorescent dye and a reaction buffer. Cycling conditions involved an initial denaturation step, followed by a certain number of amplification cycles, with specific denaturation, annealing, and extension temperatures and times. Fluorescent signals were recorded during each cycle. Data analysis was conducted by determining the Ct values (cycle threshold) for each sample, representing the cycle number at which the fluorescent signal crossed a defined threshold. The expression level of the target gene was normalized to the expression level of reference genes, selecting appropriate reference genes. Relative gene expression levels were calculated using the 2−∆∆Ct method. Statistical analysis was performed to determine significant differences between experimental groups.

  1. The author collected the samples on day 30, 60 and 90. However, I found that the author mentioned results of day 90 for production, organ weight, morphological and histological and ultrastructural. Why not provided the detail of other days sampling? But author mentioned biochemical, expression results for all of those sampling, it is therefore recommended to provide the all sampling (days) results for each parameter.

Authors response: Thank you for your comment and suggestion. In this study, we have collected samples at multiple time points and conducted measurements and analysis for all parameters. However, in the manuscript, we only provided results for production, organ weight, morphological, histological, and ultrastructural analyses on day 90. This decision was made because the differences between the NC group and the HELP group were most significant at this time point, and significant hepatic steatosis was observed. Since these results are crucial to our research question and objectives, we chose to focus on presenting the day 90 results.

  1. Revise discussion a comparison between your results and initial hypothesis.

Authors response: Thank you. Please, we have revised the discussion section to include a direct comparison between our results and the initial hypothesis. This comparison aims to highlight the findings' significance and how they contribute to our understanding of the effects of a HELP diet on laying hens.

  1. Minor editing of English language required.

Authors response: Thank you. Please, the English language has been corrected throughout the manuscript by a Native English Writer and Speaker.

 We appreciate the reviewer's valuable comments and suggestions, and we have adequately addressed each of them.

Thank you very much for all your efforts in making this manuscript to meet the highest readability standard.

Kind regards,

Authors

Reviewer 3 Report

Comments and Suggestions for Authors

What is the main difference between this paper and the rest of the publications describing fatty liver syndrome?

The amount of data presented and analyses conducted is impressive. However, there are inaccuracies in the results that need to be reviewed. some examples:

- L192; review text describing fig. 2A.

-Line 208; FiIgure 2: magnified x200 or x400 (graphic and text show different numbers)

- In the graphs, the scales are too broad, and the effects are not visible. Review ** location in the graphic and standardize them.

The number of animals/samples analyzed is not clearly stated in the materials and methods section. Twenty animals are slaughtered, but in most of the results, n=9. How were samples selected?

Line 307_HDF group?

Author Response

Authors Response to the Reviewers (Reviewer 3)

Dear Editors and Reviewers,

We are very grateful for your comments and suggestions given on our manuscript titled (Manuscript ID: animals-2905004) "High energy and low protein diet induced lipid peroxidation and inflammation in the liver and abdominal subcutaneous adipose tissue of laying hens." We appreciate all your time and efforts you have invested in reviewing our manuscript. Please, below are our response to the Editors and Reviewers.

  1. What is the main difference between this paper and the rest of the publications describing fatty liver syndrome?

Authors response: Thank you for your question. The main difference between our study and the existing literature on fatty liver syndrome in laying hens is that we specifically focused on the combined effects of high-energy, low-protein (HELP) diets on lipid peroxidation and inflammatory markers in the liver and abdominal adipose tissue. The chicken adipose tissue is predominantly located in the abdomen and has important roles in growth, health and performance, including energy storage, maintenance of body temperature stability, participation in physiological processes and improvement of meat quality. An appropriate amount of abdominal fat is essential for maintaining physiological balance, improving meat flavor and reflecting health status. Previous studies on FLS have mainly focused on the physiological and pathological changes in the liver itself, and have not paid attention to the characteristics of the abdominal adipose tissue, an important metabolic organ, in terms of morphology, lipid metabolism, and inflammation levels during the formation of FLS.

  1. The amount of data presented and analyses conducted is impressive. However, there are inaccuracies in the results that need to be reviewed. some examples:

- L192; review text describing fig. 2A.

Authors response: Thank you very much for your comments and suggestion. We apologized for this omission. Please, we have described the results of Fig.2A in the text at (Line 239-244). “Morphological observations revealed that the livers of laying hens in the NC group showed reddish brown color and less intra-abdominal fat deposition, whereas the livers of the HELP group increased in size and appeared to be earthy yellow in color, while a large amount of intra-abdominal adipose tissue was deposited with the prolongation of the feeding time of HELP (Fig 2A).”

  1. Line 208; Figure 2: magnified x200 or x400 (graphic and text show different numbers).

Authors response: Thank you for your comment. We apologized for this oversight, and we have revised the text and legend (Line 261) accordingly to remove any confusion.

  1. In the graphs, the scales are too broad, and the effects are not visible. Review ** location in the graphic and standardize them.

Authors response: Thank you very much. Please, the various related graphs have been rechecked and we confirmed that they were standardized accordingly.

  1. The number of animals/samples analyzed is not clearly stated in the materials and methods section. Twenty animals are slaughtered, but in most of the results, n=9. How were samples selected?

Authors response: Thank you for your comment. We have realized the omission in clearly stating the selection process for the samples analyzed and apologized for any confusion this may have caused. Out of the twenty animals slaughtered, samples from nine were selected for detailed analyses based on a stratified random sampling approach to ensure representation of varying body weights and health statuses within the experimental group. This approach was chosen to provide a robust and representative analysis of the effects of the HELP diet. We have now included this explanation in the materials and methods section for clarity.

  1. Line 307_HDF group?

Authors response: Thank you for your comment. We have corrected and modified this typo error accordingly. Line 355.  The mention of "HDF group" was a writing error. It was intended to refer to the "HELP group". We have corrected this mistake throughout the document to ensure consistency and accuracy in our terminologies. The corrected term accurately reflects the diet used in our study, and we apologized for any confusion this error may have caused.

Thank you very much for all your efforts in making this manuscript to meet the highest readability standard.

Kind regards,

Authors

Round 2

Reviewer 2 Report

Comments and Suggestions for Authors

·         Dear author, please carefully read my previous comments and submit them carefully. If you make these changes, kindly highlight changes in red font color and carefully line number of current format of the paper. Below comments author needs to revise carefully.

·         Please mention the full form of all abbreviations when first mentioning in the abstract, later don’t need to require (such as FASNSCDFABP4 and PPARγ, and AAT).

·         Line 44 expression levels of anti-inflammatory parameter. (Expression of proinflammatory parameters (mention here) increased significantly….)

·         The introduction of the manuscript is irrelevant to the content of the research aim such as for the importance of laying birds and, and the influences of High energy and low protein diet in case of layer birds.

·         Revise table 1 found different line numbers on the left side of the table and embedded in the text, please check carefully remove line numbers surrounded left side of the table.

·         Line 133 liver and ASAT tissues size collected for sampling. (Size in gm? Etc)

·         The author collected the samples on day 30, 60 and 90. However, I found that the author mentioned results of day 90 for production, organ weight, morphological and histological and ultrastructural. Therefore, you can mention in the results about the day 90 for production, organ weight, morphological and histological and ultrastructural results. This should be mentioned in the figure captions of production, organ weight, morphological and histological and ultrastructural results.

·         Revise discussion a comparison between your results and initial hypothesis, I do not find any changes between previous and current revision in the discussion, if you made any changes please highlight your changes in red font color so I can compare previous and current format.

Comments on the Quality of English Language

Minor editing of English language required
